# Recent Advances in Continuous MOF Membranes for Gas Separation and Pervaporation

**DOI:** 10.3390/membranes12121205

**Published:** 2022-11-29

**Authors:** Xiao Xu, Yusak Hartanto, Jie Zheng, Patricia Luis

**Affiliations:** 1Materials and Process Engineering (iMMC-IMAP), UCLouvain, Place Sainte Barbe 2, 1348 Louvain-la-Neuve, Belgium; 2Research and Innovation Centre for Process Engineering (ReCIPE), Place Sainte Barbe 2, bte L5.02.02, 1348 Louvain-la-Neuve, Belgium; 3School of Chemistry and Chemical Engineering, Chongqing University, No. 55 Daxuecheng South Rd., Shapingba, Chongqing 401331, China

**Keywords:** metal–organic frameworks, continuous MOF membranes, formation developments, gas separation, pervaporation

## Abstract

Metal–organic frameworks (MOFs), a sub-group of porous crystalline materials, have been receiving increasing attention for gas separation and pervaporation because of their high thermal and chemical stability, narrow window sizes, as well as tuneable structural, physical, and chemical properties. In this review, we comprehensively discuss developments in the formation of continuous MOF membranes for gas separation and pervaporation. Additionally, the application performance of continuous MOF membranes in gas separation and pervaporation are analysed. Lastly, some perspectives for the future application of continuous MOF membranes for gas separation and pervaporation are given.

## 1. Introduction

The demand for sustainable and energy-efficient separation and purification technology is growing in order to achieve carbon neutrality. Compared with other traditional separation processes (including amine adsorption, pressure swing adsorption, and cryogenic separation), membrane separation has gained increasing attention from different realms, namely from energy, water, chemical, healthcare, and food areas, because it has unique merits, such as energy efficiency, high separation efficiency, small carbon footprint, low cost, easy operation, and scalability, etc. [1,2,3]. Among a variety of membranes, semi-permeable membranes are central in gas separation and pervaporation to separate or remove angstrom-scale gas or liquid molecules [4]. Gas separation is considered as one of the most promising technologies to perform continuous, reagent-free, and non-hazardous separation of gases or to obtain mixtures enriched in one component [5,6]. Pervaporation is also an important membrane technology that is typically used to separate minor components from heat-sensitive, close-boiling, and azeotropic mixtures. Compared to traditional distillation, pervaporation is a benign process with high efficiency [7].

Metal–organic frameworks (MOFs) have gained more attention for gas separation and pervaporation because MOFs are a class of porous materials with unique traits, including large surface areas, uniformly adjustable sized pores, high void volumes, and hybrid organic–metal nature [8,9,10,11,12]. Membranes based on these kinds of materials generally have an asymmetric structure with a selective layer positioned on inorganic supports [13]. Up to now, gas separation and pervaporation using continuous MOF membranes are mainly achieved by exploiting the solution–diffusion mechanism of the MOF. Specifically, the solution–diffusion mechanism can be classified into two separation mechanisms: solubility-based separation and diffusivity-based separation [14].

Although there have been several excellent review articles on the application of MOF membranes for gas separation and pervaporation [8,9,15], only continuous MOF membranes applied in gas separation or pervaporation have been summarized. In this work, recent developments, including the fabrications and applications, of continuous MOF membranes for gas separation or pervaporation are systematically summarized and analysed. Moreover, the application performance of continuous MOF membranes for gas separation and pervaporation are evaluated and analysed, respectively. In addition, the major challenges and future perspectives of continuous MOF membranes for gas separation and pervaporation are presented.

## 2. Developments of the MOF-Based Membrane

As a type of porous material, metal–organic frameworks (MOFs), because of large surface areas, good stability and well-defined pore structures with molecule-scale pore size, show great potential for energy-efficient separations of gas or liquid mixtures [16,17]. Because most MOFs are in the form of either microcrystalline powders or submillimetre crystals, it is difficult to use them widely in the field of separation. To improve the situation, a variety of MOF-based membranes have been prepared. These membranes can in general be classified into two types: continuous MOF-based membranes and MOF-based mixed matrix membranes. In this review, only continuous MOF-membranes are discussed.

### 2.1. Continuous MOF-Based Membranes

Continuous MOF-based membranes are a class of continuously grown membranes which are mainly prepared by growing uninterrupted polycrystalline or epitaxial layers on substrates. The substrates to support selective MOF layers are typically inorganic. Some efforts have been made to deposit MOF layers on organic substrates over these years [18,19]. In this type of membrane, continuous MOF layers have selectivity to different feed components because the pores of MOFs are the only permeation channel through all layers, while substrates as a support offer mechanical strength for the MOF layers. Since the first attempt to make continuous MOF membranes was reported by Fischer et al. in 2005 [20], there has already been a variety of approaches developed to fabricate continuous MOF films, including in situ growth [21,22,23], secondary growth [24,25,26], interfacial diffusion [27,28,29,30,31], liquid phase epitaxy [32,33,34,35], vapor deposition [36,37,38,39,40,41], and the electrodeposition method [42,43,44].

#### 2.1.1. In Situ Growth

In situ growth is to fabricate continuous MOF-based membranes on substrates by soaking the substrate in solutions of inorganic precursors/organic precursors, as illustrated by Figure 1. The whole process is mainly divided into two steps: the first step is to induce MOF crystal nucleation on the substrate surface under certain conditions, and the second step is to promote the continuous growth of the MOF crystals until the substrate surface is completely covered with a continuous MOF layer [45].

However, it is difficult to achieve sufficient density of nucleation sites on unmodified substrates. To solve this difficulty and at the same time strengthen the adhesion MOF layers and substrates, the supporting substrates are modified. A modification of the substrate surface is sometimes crucial to form crack-free MOF membranes. In summary, in situ growth can be further classified into two routes: in situ growth on an unmodified support and in situ growth on a modified support [9]. Despite many attempts, the fabrication of a defect-free MOF layer on an unmodified substrate remains a challenge. To obtain a defect-free continuous MOF layer while enhancing the adhesion of the MOF layer and the substrate, the substrate usually needs to be modified.

As the name suggests, in situ growth of the MOF on an unmodified support is to grow a MOF layer on an unmodified support. The first example of a continuous and well-intergrown MOF-5 membrane was demonstrated by Lai and co-workers [46]. A porous alumina disc was soaked into the mother solution of MOF-5, and then the solution was heated to 105 °C under sealed conditions to prepare continuous MOF-5 membranes. The tested results showed that the diffusion of H_2_, CH_4_, N_2_, CO_2_, and SF_6_ through the MOF-5 membrane followed the Knudsen diffusion behaviour. It was reported by Jürgen et al. that by replacing dimethylformamide (DMF) as the solvent and aqueous methanol with absolute methanol, a crack-free, dense and polycrystalline ZIF-8 layer was synthesized on a titania support using an in-situ microwave-assisted solvothermal method [47]. The as-synthesized MOF membrane showed a fine balance between permeance and selectivity for the separation of H_2_/CH_4_. To enhance the adhesion between the MOF layer and the substrate, Chen and Wang et al. demonstrated the in-situ growth of ZIF-8 layers on tubular substrates using a concentrated synthesis gel, with a molar composition of Zn, 2-methylimidazole (Hmim), and methanol of 1:8:700, by dip or slip casting [48]. The hollow substrates were first dried at 80 °C overnight and then being placed vertically in an autoclave. Subsequently, the ZIF-8 synthesis solution was slowly poured into the autoclave and heated under the given conditions. The crystalline ZIF-8 layers on the substrates were compact and continuous with a thickness of 6 μm. The EDX elemental mapping suggested that some ZIF-8 crystal might exist in the pores of the supports. The prepared ZIF-8 exhibited strong CO_2_ adsorption properties. For the enhancement of Interfacial adhesion, Kang and co-workers used ‘single metal source’ to generate a homochiral MOF membrane [49]. In this report, the substrate not only played the role of supporting the MOF layer but was also involved in the MOF growth process as the sole nickel source. The obtained thin and crack-free Ni_2_(L-asp)_2_(bipy) exhibited a comparable ee value of 32.5% for the pervaporation separation of pure racemic mixtures.

As noted above, how to enhance the interfacial adhesion is an urgent problem for the preparation of continuous MOF membranes; the modification of the substrate has been considered as an effective strategy to solve this problem. The substrate surface can be modified physically or chemically. Zhang et al. first reported the growth of low-defect ZIF-8 membranes on tubular α-alumina substrates that were modified by coating a layer of ZnO, and then 2-methylimidazole was used to activate and react with the ZnO layers [50]. The resulting ZIF-8 layers on physically modified supports exhibited excellent separation performance. These ZnO layers have an important role in promoting the synthesis of the ZIF-8 membranes because ZnO can be utilized as the secondary metal source for the preparation of MOF membranes. The obtained low-defect membrane exhibited higher permeances for small gas molecules of H_2_, N_2_, and CH_4_, while large molecules of C_3_H_8_, n-C_4_H_10_, and SF_6_ had significantly low permeability through the ZIF-8 membranes. For the preparation of defect-free MOF membranes, Tan and co-workers prepared thin uniform ZIF layers on physically modified supports [51]. First, they covered the surface of polypropylene (PP) supports with the metal-chelating polyaniline (PANI) layers by an in situ deposition technique. Subsequently, ZIF-7 and ZIF-67 membranes with controllable thicknesses were produced on the PP supports. The ZIF-7-NH_2_ membrane with gate-opening and flexibility showed H_2_ permeance of 4.9 × 10^−6^ mol m^−2^ s^−1^ Pa^−1^ and an ideal H_2_/CO_2_ separation factor of 10.6. Using the physically modified concept, Wu et al. deposited 50–200 nm thick polydopamine (PDA) layers on the α-Al_2_O_3_ disks before the in situ solvothermal synthesis of MIL-160 membranes [52]. The obtained high-crystallinity MIL-160 membrane showed the high flux of 467 g m^−2^ h^−1^ and the selectivity of 38.5 from the pervaporation separation of p-xylene from its bulkier isomer o-xylene. Another example of the preparation of HKUST-1 membrane on the chemically modified support via the ‘twin copper source’ technique was reported by Guo and co-workers [53]. In their report, the copper net as both copper source and support for the continuous MOF membrane first went through a series of treatments to reach oxidation, and then the HKUST-1 membrane was synthesized on the oxidised support that was placed vertically in an autoclave full of a solution of HKUST-1. The resulting membrane demonstrated higher permeation flux of H_2_ (~1 × 10^−1^ mol·m^−2^·s^−1^) than other gas components in the separation of H_2_/CO_2_, H_2_/N_2_, and H_2_/CH_4_ mixtures.

#### 2.1.2. Secondary Growth

Secondary growth (also called seeded-secondary growth) of MOF membranes involves the seeding of a crystal nucleus on the support, followed by the growth of MOF membranes on the seeded support, as illustrated in Figure 2. For the fabrication of MOF membranes, secondary growth has been proved to be more likely to obtain thinner and less defective MOF membranes [54]. Although the crystal nucleation and growth steps of secondary growth can be independently controlled, the control of the final thickness and orientation of MOF membranes can be achieved by controlling the thickness and orientation of the seeded layer [55]. The key step for the secondary growth of MOF membranes is the seeding. Up to now, a variety of seeding techniques has been developed including rubbing [56], dip-coating and slip-coating [57], electro-spinning [58], microwave radiation [59], spin coating [60], reactive seeding [61], and thermal seeding [62], which can be classified into physical seeding and chemical seeding.

In 2009, Tsapatsis et al. synthesized the microporous MOF membrane by manual assembly using a seeded solvothermal growth on an α-alumina support coated with polyethyleneimine (PEI) [63]. In this method, the seeded support was vertically placed in the autoclave containing a mother solution of microporous MOF membrane at 150 °C for 12 h, producing an oriented membrane with the pores aligned preferentially perpendicular to the support surface. Such MOF membranes exhibited low H_2_ permeances with ideal selectivity for H_2_/N_2_ of 23. Lai and co-workers also reported on the preparation of a continuous ZIF-8 membrane [64]. The seeds were first attached on a hollow yttria-stabilized zirconia (YSZ) fibre support by dip-coating and drying, and then the seed support was placed vertically in a glass vial filled with a secondary solution. Subsequently, the glass vial was heated under certain conditions, finally producing a continuous ZIF membrane with a thickness of about 2 μm. In single-component permeation studies, the membrane exhibited a superior performance with H_2_ permeance of ~15 × 10^−7^ mol/m^2^ s Pa and an ideal selectivity of H_2_/C_3_H_8_ of more than 1000 at room temperature. Wang et al. used a wet-rubbing method to coat Ni_2_(l-asp)_2_bipy crystals on the top surfaces of SiO_2_ discs [65]. After this operation, a solvothermal reaction was carried out to prepared continuous grown Ni_2_(l-asp)_2_bipy membrane. To apply this membrane for the pervaporation separation of ethanol from ethanol/water mixture, a PDMS layer was subsequently coated on a Ni_2_(l-asp)_2_bipy layer to switch the wettability of the MOF membrane to the hydrophobicity. The prepared membrane was used to separate the water/ethanol mixture with 50 wt% ethanol at 30 °C, exhibiting a remarkably high flux of 27.6 kg·m^−2^ h^−1^ and a separation factor of 73.6. For high reproducibility, Jürgen Caro’s group used an automatic dip-coating device to attach ZIF-8 seeds on α-Al_2_O_3_ supports before the secondary growth in solvothermal conditions [66]. Supported polycrystalline ZIF-8 membranes still did not display a clear cut-off in the pervaporation separation of n-hexane, benzene, and mesitylene.

**Figure 2 membranes-12-01205-f002:**
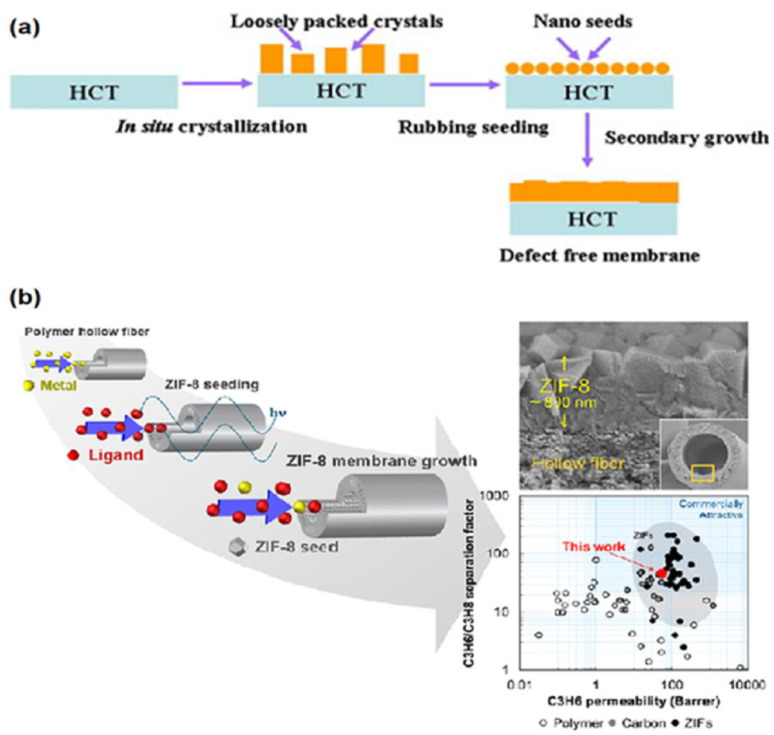
Schematic representations of the secondary growth of MOF membranes: (**a**) physical seeding and (**b**) chemical seeding. Reprinted with permissions from Refs. [67,68].

In addition to physical seeding, chemical seeding has been gradually introduced to deposit MOF crystals on the porous substrates. “Thermal seeding” was first reported to anchor MOF seeds on porous substrates by Jeong et al. [62]. In this research, the solution containing HKUST-1 crystals and unreacted HKUST-1 precursors was dropped on hot porous α-alumina supports, and this process repeated until the support surface was covered sufficiently with seed crystals. The resulting seeded support was placed in the autoclave, into which the solution containing HKUST-1 precursors was then filled. Under appropriate synthesis conditions, a continuous HKUST-1 membrane without cracks and fractures grew on the chemically seeded support. The prepared HKUST-1 membrane showed moderate ideal selectivity of H_2_ over N_2_, CH_4_, and CO_2_ for the gas separation, also exhibiting greater affinity to CO_2_ over CH_4_ and N_2_ with the temperature. Another chemically seeding method is facile reactive seeding (RS) [61]. In this method, a porous substrate played dual roles, one as the support to hold the selective layer, and the other as the inorganic source to engage in the growth of MIL-53 seeds on the support. The as-synthesized MIL-53 membrane displayed a sharp separation for the pervaporation separation of H_2_O from ethanol, t-butanol, and ethyl acetate, respectively. At 60 °C, when a H_2_O-ethyl acetate solution (7 wt.% H_2_O) passes through the MIL-53 membrane, the H_2_O concentration in the permeate increased to 99 wt.% with the flux of 454 g·m^−2^ h^−1^.

Lee et al. used a combination of microwave-assisted seeding and microfluidic secondary growth to fabricate well-intergrown ultrathin ZIF-8 membranes with a thickness of ~800 nm deposited on porous Matrimid^®^ polymer hollow fibres, which was the first time that the ZIF-8 membranes via secondary growth were grown on the bore side of polymeric hollow fibres [68]. They tested binary gas separation performance of the membranes using propylene/propane, showing a separation factor of about 46 and propylene permeance of ~55 GPU (gas permeation unit, 1 GPU = 3.348 × 10^−10^ mol m^−2^ s^−1^ Pa^−1^). Using a covalent-assisted seeding method, well-intergrown ZIF-8 membranes were prepared on polyimide substrate [69]. For the successful fabrication of a MOF layer on the polymeric substrate, the imidazole-2-carbaldehyde (ICA) molecules as the covalent agent were first grafted onto the polyimide substrate cross-linked with ethylenediamine (EDA), which offered the reaction sites for anchoring ZIF-8 seeds on polyimide substrate. By this method, a well-grown ZIF-8 layer was firmly coated on the polyimide substrate. The prepared ZIF-8 membrane showed a further elevated separation factor compared to polyimide membrane for the pervaporation dehydration of isopropanol at 40 °C.

#### 2.1.3. Interfacial Synthesis

So far, various methods of interface synthesis have been developed and used successfully in the facile manufacture of MOF membranes. According to the fluid properties of the precursor solution, interface synthesis can be divided into static and dynamic interface synthesis methods (Figure 3) [70].

##### Static Interfacial Synthesis

Static interfacial synthesis used for obtaining continuous MOF layers can be classified into the following two routes: (1) interfacial diffusion synthesis [71], (2) contra-diffusion synthesis (counter-diffusion synthesis), as shown in Figure 3a,b [72]. In the interfacial diffusion synthesis, two precursors (e.g., the metal ion and the organic ligand) are usually separately dissolved in two immiscible solvents. When two precursor molecules independently meet at the interface in two incompatible solvents, the crystallization that only occurs at the liquid/liquid or liquid/solid interface leads to the formation of the MOF membrane. Similarly, contra-diffusion synthesis has two precursor solutions as well. It differs from the former route in that the two precursor solutions in contra-diffusion synthesis are separated by a porous substrate; two precursor molecules diffuse in opposite directions through the porous channels of the substrate.

Zhu et al. used the interfacial diffusion method to synthesize free-standing continuous MOF membranes [73]. To grow the MOF membrane, the reactant concentration region needed to be identified because a MOF layer was only able to be prepared in the region of high molar ratio of Zn(NO_3_)_2_ to terephthalic acid (H_2_BDC) and low triethylamine (TEA) concentrations, in which TEA was utilized as a catalyst to improve the coordination reaction between Zn(NO_3_)_2_ and H_2_BDC. The resulting membranes were found to be asymmetric including three layers by SEM and XRD, the composition of the surface layer was 3D Zn_4_O(BDC)_3_ (MOF-5) with a particulate morphology, the bottom layer had a sheet-like morphology, and its component was 2D ZnBDC·DMF (MOF-2), and in between were transitional zinc-carboxylate structures, viz. Zn_3_(OH)_2_(BDC)_2_ (MOF-69c) and Zn_5_(OH)4(BDC)_3_. Due to poor mechanical stability, free-standing continuous MOF membranes have not been widely used in gas separation and pervaporation [9]. Chen et al. prepared a ZIF-8 membrane on porous α-Al_2_O_3_ disks via interfacial synthesis [74]. In this method, the pre-treated α-Al_2_O_3_ disk was soaking into inorganic precursor solution in an Erlenmeyer flask. To fully fill the pores of the disk with zinc nitrate, the flask was connected to a vacuum. After 30 min, the disk was taken out, and then the excess solution on the disk surface was wiped out using a rubber wiper. Subsequently, the disk was gently soaked into the pre-heated organic precursor solution for the desired time to prepare the ZIF-8 membrane. The obtained ZIF-8 membrane exhibited a CO_2_/N_2_ separation factor of 5.49 and CO_2_ permeance of 0.47 × 10^−7^ mol·m^−2^·s^−1^·Pa^−1^ in single-gas permeation experiments of CO_2_ and N_2_.

Like interfacial diffusion synthesis, the contra-diffusion method has been developed to construct ultrathin MOF membranes. Wang and co-workers successfully prepared ZIF-8 films on flat nylon substrates with the contra-diffusion synthesis method in 2011 [72]. In their research, the zinc nitrate methanol solution and 2-methylimidazole (Hmim) methanol solution were separated by the porous nylon membrane with a pore size of 0.1 μm. After a certain period of crystallization, the ZIF-8 films were prepared under ambient temperature and deployed to separate the H_2_/N_2_ gas mixture. The gas permeation results of a H_2_/N_2_ ideal selectivity of 4.3 with H_2_ permeance of 1.97 × 10^−6^ mol m^−2^ s^−1^ Pa^−1^ and N_2_ permeance of 0.46 × 10^−6^ mol m^−2^ s^−1^ Pa^−1^ confirmed that the ZIF-8 layers were continuous and compact. In 2013, the same group extended the contra-diffusion synthesis method to synthesize continuous and compact ZIF-8 layers on nylon membranes by using an aqueous solution with a Zn^2+^:Hmim stoichiometric ratio of 1:2 [75]. A 2.5 μm thick ZIF-8 layer was formed on the hmim side, exhibiting a H_2_/N_2_ ideal selectivity of 4.6 in the gas permeation tests.

**Figure 3 membranes-12-01205-f003:**
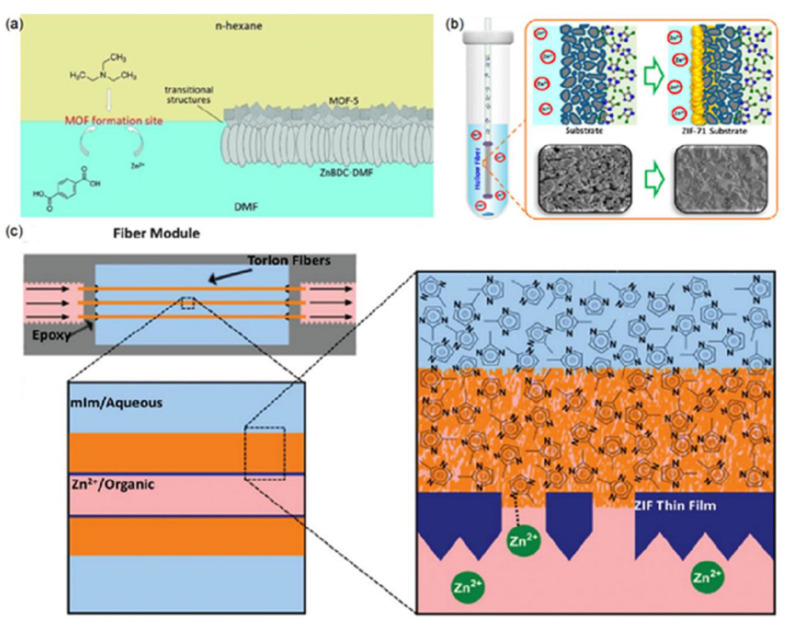
Schematic representations of the interfacial synthesis of MOF membranes (**a**) interfacial diffusion synthesis [64], (**b**) contra-diffusion and (**c**) dynamic interfacial synthesis. Reprinted with permissions with Refs. [73,76,77].

In the contra-diffusion reaction, the substrates used as porous supports were usually commercial microfiltration membranes with large pores (pore diameter > 100 nm) and long channels (over tens of microns). In these cases, the diffusions of metal ions and organic ligands inside these membranes were quite fast and difficult to control, which resulted in the formation of MOF films with defects and/or relatively thick layers [60,78,79,80]. Considering such problems, a method of fabricating ultra-thin and continuous MOF membranes, a spatially confined contra-diffusion synthesis process, was reported by Zhang and co-workers [80]. In this process, the difference from the previously reported contra-diffusion processes was that an ultra-thin nano-porous membrane made of polydopamine-coated single-wall carbon nanotube (PD/SWCNT) network with small pores or channels of 5–10 nm was deployed as the porous substrate at the liquid–liquid interface instead of commercial microfiltration membranes. By a spatially confined contra-diffusion process, an ultra-thin continuous ZIF-8 film with a thickness of about 550 nm was formed. When applied to hydrogen separation, the membrane showed a high H_2_ permeance of up to 6.31 × 10^−7^ mol·m^−2^·s ^−1^∙pa^−1^, while maintaining a high ideal selectivity of 43 for H_2_/CO_2_. The contra-diffusion method was also utilized to fabricate a ZIF-71 membrane for the pervaporation recovery of ethanol from H_2_O [76]. At 25 °C, the synthesized ZIF-71 hollow fibre membrane displayed a comparable performance with a flux of 2.601 g/m^2^ h and a separation factor of 6.88.

##### Dynamic Interfacial Synthesis

Dynamic interfacial synthesis mainly refers to interfacial microfluidic membrane processing (IMMP). Prior to IMMP, microfluidic processing was deployed to grow continuous MOF membranes on the substrates by flowing mixed precursors in the hollow fibre. Huang and co-workers successfully deposited the ZIF-8 film with a thickness of 2.0 μm on the inner surface of a ceramic hollow fibre by microfluidic processing [81]. The as-synthesized ZIF-8 film on the inner-side of the ceramic hollow fibre was used to recover H_2_, displaying good performance with a H_2_ permeance of 4.324 × 10^−7^ mol^−2^s^−1^Pa^−1^ and H_2_/CH_4_ ideal selectivity of 12.13. Using a similar method, the ZIF-93 membrane [82] and the defect-free ZIF-8 membrane [68] were fabricated on the inner surface of the co-polyimide P84 hollow fibre and the Matrimid hollow fibre, respectively. To avoid the occurrence of bulk MOF crystallization during microfluidic processing, Nair and Jones et al., combining microfluidic processing and interface synthesis, proposed interfacial microfluidic membrane processing (IMMP) for the production of ZIF-8 membranes on torlon hollow fibres [77]. Distinguished from the microfluidic synthesis with mixed precursor solution, a Zn^2+^/1-octanol solution was passed through the bore of the hollow fibre, while a 2-methylimidazole (2-mIm)/H_2_O solution maintained stationary on the shell side. A continuous and thin ZIF-8 membrane was synthesized on the inner surface of the hollow fibre, due to the relatively fast diffusion of reactants to the water-oil interface under continuous flow. By contrast, under static growth conditions, dense and noncontinuous layers of ZIF-8 particles were produced owing to the insufficient supply of Zn^2+^ ions to maintain the growth of the film after the initial nucleation and growth of ZIF-8 crystals on the inner surface of the fibre. The prepared ZIF-8 membranes with a thickness of 9 μm were used to distinguish H_2_ from C_3_H_8_ and C_3_H_6_, exhibiting moderate and efficient selectivity. From the above example, we can conclude that the so-called IMMP involves the combination of membrane module construction and the crystallization mechanism of interface diffusion synthesis or contra-diffusion synthesis. The two non-mixed precursors flow on the tube and shell sides of the hollow fibre respectively. The MOF film is deposited on the surface of the hollow fibre by the flowing precursors. Employing the same concept, Marti et al. synthesized continuous defect-free ZIF-8 thin films on the outer surface of polymeric hollow fibre using continuous precursor flow, in which a Zn^2+^/H_2_O solution was flowed on the shell side of hollow fibre, and at the same time 2-methylimidazole (MeIm)/H_2_O was passed through the bore of the hollow fibre [83]. The prepared ZIF-8 membrane was applied to capture CO_2_, demonstrating a CO_2_ permeability of 22 GPUs and a highest reported CO_2_/N_2_ selectivity of 52.

#### 2.1.4. Liquid Phase Epitaxy (LPE)

For liquid-phase epitaxy (LPE, also known as layer-by-layer) growth, also referred to stepwise layer-by-layer (LBL) assembly, a substrate is exposed to one type of MOF precursor component (typically two types) at a time, and repeatedly cycled between organic secondary building units (SBUs) and inorganic SBUs by manual immersion [84], dipping [85], spraying [86], or spin coating [87]. Unreacted components are usually removed by rinsing a solvent between successive deposition steps, as illustrated in Figure 4. The orientation and the thickness of the MOF membrane in the vertical direction to the substrate (out-of-plane) can be precisely controlled by LPE. Moreover, the sequential and stepwise method gives each kind of precursor component the opportunity to saturate all the possible deposition sites without forming new nucleus on the surface or in the solution, thereby forming a low-defect membrane. The layer-by-layer growth mode for the fabrication of MOF membranes can be traced to the pioneering work of Wöll and co-workers [88]. They grew a MOF layer on COOH-terminated self-assembly monolayers (SAM) using a layer-by-layer approach in which SAM was repeatedly manually immersed in a solution of metal precursor and in the organic precursor solution. While there is no doubt that the manual immersion technique used in LPE applications can result in smoother MOF membranes, called surfaced-attached MOFs (SURMOFs), there is one major drawback: it is not efficient to prepare SURMOF membranes. Some improved strategies, based on the manual liquid-phase epitaxy synthesis route, are emerging to design SURMOF membranes on the substrate. For example, an automated, computer-controlled dipping robot has been deployed for the preparation of SURMOF membranes of ZIF-8 on Au-coated α-Al_2_O_3_ supports by Valadez Sánchez and co-workers [89]. The obtained ZIF-8 SURMOF membranes exhibited higher hydrogen and ethene permeances due to their smaller molecular sizes.

**Figure 4 membranes-12-01205-f004:**
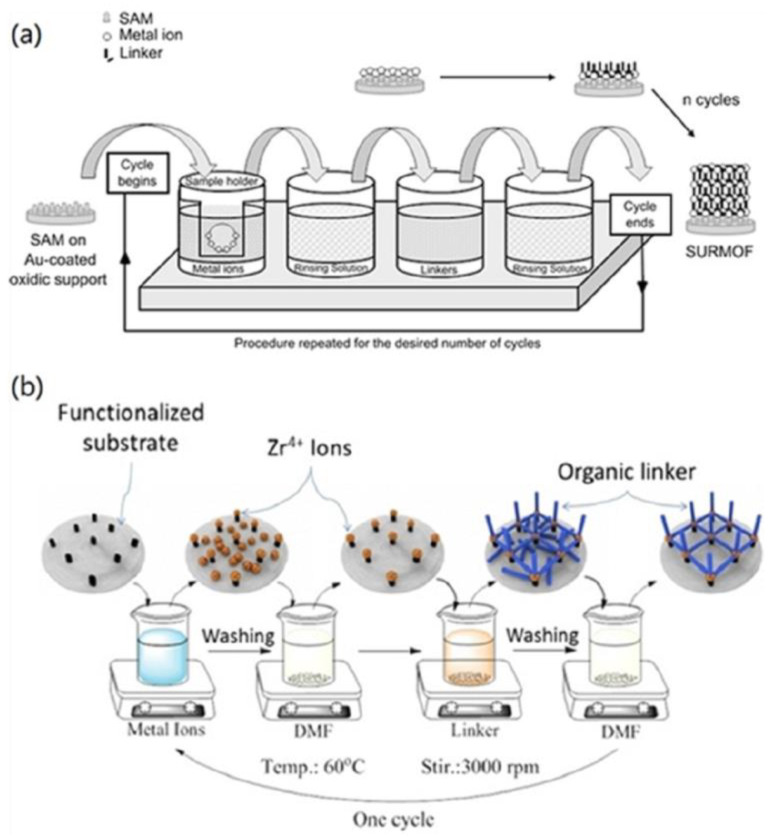
Schematic diagrams for the LPE growth of MOF membranes: (**a**) ZIF-8 SURMOF membrane deposited on a Au coated alumina support and (**b**) UiO-66-NH_2_ SURMOF membrane deposited on a flat gold-coated substrate. Reprinted with permissions from Refs. [89,90].

In 2014, Eddaoudi and co-workers reported the assembly of ultra-thin and defect-free ZIF-8 selective layers on porous aluminate supports by precisely controlling the epitaxial growth [85]. In the single-gas permeation tests of He, H_2_, CO_2_, O_2_, N_2_, CH_4_, C_2_H_4_, C_2_H_6_, C_3_H_6_, C_3_H_8_, and n-C_4_H_10_, the resultant ZIF-8 membranes were observed to have a cut-off for the diffusion coefficient between C_2_ and C_3_ because of the relatively small window size of ZIF-8. Although the ZIF-8 membranes fabricated by LPE exhibited about 100 times lower permeance but with good ideal selectivity of 5, 11, 12, and 60 for H_2_-CO_2_, H_2_-N_2_, H_2_-CH_4_, and H_2_-C_3_H_8_, respectively, at 308 K, suggesting the synthesized ZIF-8 membranes with minimal grain boundary micro-defects.

Recently, the Léon group used the LPE method and an in situ growth method to prepare an UiO-66-NH_2_ SURMOF membrane and an UiO-66-NH_2_ membrane, respectively [91]. In the equimolar case, the UiO-66-NH_2_ SURMOF membrane exhibited a higher H_2_/N_2_ separation factor of 3.02 because of the LPE method continuously generating a highly oriented MOF membrane.

#### 2.1.5. Vapor Deposition

For the MOF membranes by chemical solution deposition (CSD) typically involving reacting metal salts and organic ligands in solutions, the advantages provided are generally simplicity and effectiveness, low capital cost of laboratory implementation, compatibility with conventional wet chemical operations, and high chemical flexibility. However, the reactant transport and fluid dynamic processes of CSD are usually complex and are very difficult to control carefully, especially on an industrial scale. In contrast to CSD, vapor deposition of MOF membranes can facilitate their implementation in lab-scale research and industries. Usually, each vapor deposition process includes three steps: (1) evaporation, (2) the transportation of the vaporized material, (3) deposition of the film on the substrate. As of now, while there are many variations of vapor deposition, the processes can typically be divided into two categories: physical vapor deposition (PVD) and chemical vapor deposition (CVD), as shown in Figure 5a,b [92].

The PVD is a purely physical process in which the target material is evaporated into the gas phase and deposited onto the substrate for the preparation of the thin films without any chemical reaction taking place. The most critical step in the entire process is the initial evaporation of the target material, which can be achieved by various methods, such as conventional thermal evaporation, ion plating, sputtering, or pulsed-laser deposition (PLD). However, it is quite difficult to apply PVD to the fabrication of the MOF due to low vapor pressure and thermal decomposition or amorphization of MOFs from thermal stress, ion bombardment, or laser irradiation [93,94]. To remedy those limitations, the femto-PLD technique was utilized to fabricate porous ZIF-8 membranes for the first time [94]. To prevent its structural degradation and decomposition, polyethylene glycol 400 (PEG-400) was deployed as a “vehicle” to impregnate with ZIF-8, and then hybrid ZIF-8 soaked with PEG-400 was ablated in an ultrahigh vacuum chamber, finally forming membranes with approximate composition Zn(C_3_N_2_H_2_–CH_3_)2·^1^/_6_PEG-400 through the femto-laser technique. The PEG additive in the membranes could be removed by washing with ethanol. However, MOF membranes prepared by PVD for gas separation and pervaporation have not reported so far.

In contrast to PVD, CVD typically involves a chemical reaction between solid and gas phases. In other words, this deposition method is based on the reaction of vaporized substances with the substrate, with the growing layer and/or with each other to form a uniform layer. Up to now, several different routes have been explored using CVD for the preparation of MOF films, mainly including atomic layer deposition (ALD) and molecular layer deposition (MLD). The ALD is a vapor-phase thin layer deposition method based on alternating surface reactions between two precursor gas pulses. This self-limiting growth mechanism well controls the conformal deposition of the layers with nanometre precision thickness [95]. The MLD is a method similar to ALD. In MLD, a layer of molecules is deposited during one self-limiting surface reaction step instead of the deposition of monolayer of atoms. In 2013, Salmi and co-workers deposited MOF-5 films from the vapor phase by ALD [96]. In their study, it was found that the as-deposited layers were amorphous and nonporous. For forming porous MOF-5 thin membranes, the as-deposited membranes also required a two-step post-deposition crystallization treatment. In 2018, the smooth, pinhole-free ZIF-8 thin layers were uniformly deposited on porous substrates using ALD [97]. In one ALD cycle, a ~5-μm γ-alumina mesoporous layer with pores in the 2 to 5 nm range was first coated on an α-Alumina macroporous substrate, then diethylzinc was deposited on the γ-alumina mesoporous layer using ALD, and the diethylzinc layer was finally exposed to water vapor to yield hydroxylated zinc oxide. This ALD cycle was repeated for up to 50 times for the preparation of an impermeable zinc oxide layer coated on γ-alumina layer. This impermeable material can be transformed into a porous ZIF-8 membrane by exposure to vapor of 2-methylimidazole. For the separation of propylene/propane, the resultant porous ZIF-8 membrane enabled a propylene/propane selectivity over 100 with a propylene permeance as high as 114 GPU.

Nilsen and co-workers reported that the film of amino-functionalized UIO-66 was deposited through an all-gas-phase ALD/MLD [98]. Both ZrCl_4_ and 2-amino-1,4-benzene dicarboxylic acid (2-amino-1,4-BDC) were utilized as precursors to perform the ALD/MLD depositions in an F-120 Sat-type ALD reactor. The as-formed films could be crystallized by post-deposition treatment. In 2020, Lausund et al. reported on MOF thin films with bi-aromatic linkers deposition by MLD [99]. Although some progress for the fabrication of MOF membranes using MLD has been made, there are no reports on the application of MOF membranes prepared via MLD for gas separation and pervaporation because MOF membranes fabricated using this method generally require a postdeposition treatment to convert amorphous MOF layers into crystalline MOF layers [100].

**Figure 5 membranes-12-01205-f005:**
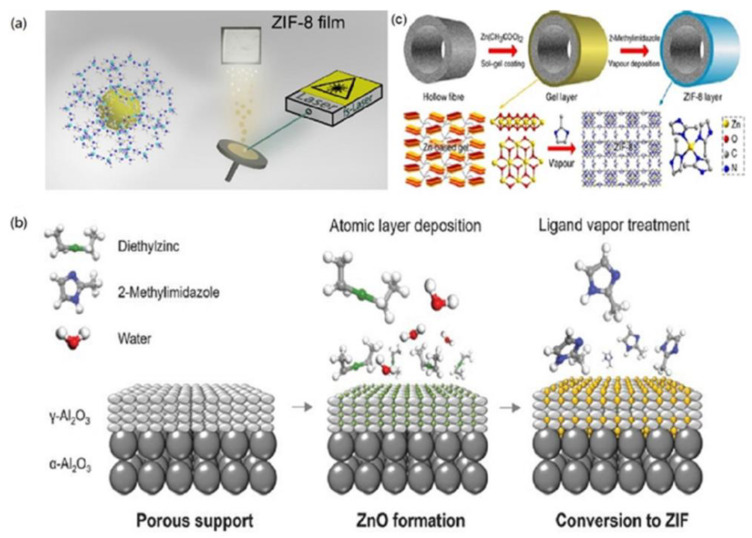
Schematic of MOF membrane formation processes using vapor: (**a**) ZIF-8 membrane deposition via PVD, (**b**) ZIF-8 membrane deposition via CVD and (**c**) ZIF-8 membrane deposition via GVD. Reprinted with permissions from Refs. [94,97,101].

To easily produce the MOF membrane with large effective membrane areas, Li and co-workers developed a gel-vapor deposition (GVD) method for preparing ultrathin ZIF-8 membranes by a combination of modification-free sol-gel coating and solvent-free vapor deposition (Figure 5c) [101]. In this method, the Zn-based gel, composed of zinc acetate dihydrate and ethanolamine, was first coated on ammoniated polyvinylidene fluoride (PVDF) hollow fibres by heat treatment. Then, the organic groups of the gel were substituted through ligand vapor deposition by heat treatment, forming the ZIF-8 membrane. Due to the smaller kinetic diameter of H_2_, H_2_ permeance through the resultant ZIF-8 membrane was up to 215.4 × 10^−7^ mol m^−2^ s^−1^ Pa^−1^ with a H_2_/C_3_H_8_ selectivity of as high as 3400.

#### 2.1.6. Electrodeposition

A major disadvantage of many of the methods described above is that defects are randomly generated during the MOF membrane processing. Although a relatively thick membrane can minimize defects, it inevitably reduces the permeability of the membrane [102]. Moreover, a large amount of waste is produced during the fabrication process, especially given the inorganic salts used and the risks and costs related to anions such as nitrate and chloride ions [103]. Electrochemical deposition stands out among the methods of preparing MOF membranes.

The electrochemical method is advantageous because it requires a relatively minor or non-pretreated substrate, shorter growth time, and milder preparation, and is easier to scale up. More importantly, this method allows for controlling the morphology and thickness by altering the current. In principle, there are three routes for making MOF thin membranes by the electrochemical method, namely anodic dissolution, cathodic deposition, and electrophoretic deposition (Figure 6) [102,104]. Anodic dissolution is based on the dissolution of a metal plate used as the anode and the inorganic source for the formation of MOF membranes, and subsequently the released metal ions react with the organic ligands in the electrolyte [103]. During the cathodic deposition process of MOF membranes, electrodes are utilized as chemically inactive spectators and only serve as a source of electrons. Hydroxide formed from the reduction reaction raises the pH of the electrolyte near the cathode and promotes the nucleation and crystal growth of MOFs on the substrate containing the respective organic and inorganic precursor by deprotonating organic ligands [43]. As for electrophoretic deposition, the charged MOF particles (possibly due to missing linkers or missing metal nodes) result in an electric field formation in the electrolyte, in practice, which drives the charged particles toward the oppositely charged electrode to form a MOF layer [105].

The HKUST-1 membrane was among the first MOF membranes to be electrochemically fabricated by researchers at BASF and is also the first MOF membrane reported that has been deposited on a substrate by anodic dissolution [106,107]. During the process, the solvent was continuously stirred, resulting in the removal of crystals formed from the anode surface, resulting in the continuous production of MOF particles. Since then, MOF layers synthesized on the anode have been used to detect nitro compound explosives [108], glucose [109], and to store charge in the electrical double layer [110], as well as in separation [111,112]. So far, the development of electrochemical MOF deposition on other conductive substrates (including indium tin oxide (ITO), fluorine tin oxide (FTO), and glassy carbon) by anodic dissolution has been far behind that achieved by cathodic deposition [113]. To open up new possibilities for producing MOF layers on transparent conductive substrates, the Oliver group confirmed using a modified scheme that a series of MOF thin layers were deposited on indium tin oxide (ITO) glass by anodic dissolution [114]. Before partial dissolution of the anode and initial nucleation of the MOF crystals, the ITO support needed to be electrocoated with either copper or zinc microcrystalline films (Cu-ITO and Zn-ITO), respectively. These metallic coatings served as the metal cation source to grow HKUST-1 and the two-dimensional (2D) cationic framework [Cu(C_10_H_8_N_2_)^2+^]Br_2_ (CBBr) on Cu-ITO substrates, and Zn_6_(OH)_3_(BTC)_3_(H_2_O)_3_·7H_2_O (Zn-BTC, BTC = benzene tricarboxylate) and the 2D framework [Zn(BPDC)-(H_2_O)]·H_2_O (Zn-BPDC, BPDC = 2,2′-bipyridine-5,5′-dicarboxylate) on Zn-ITO substrates. In 2009, Ameloot et al. electrochemically deposited a MOF layer on an anodic surface by careful modification of the conditions used to synthesize bulk MOF material [115]. In their report, by applying an appropriate electrical voltage to the anodic electrode, the electrochemical oxidation of Cu-atoms on the anode led to the dissolution of Cu (II) ions, which were subsequently introduced into the electrolyte containing 1,3,5-benzenetricarboxylic acid (BTC) and methyltributylammonium methyl sulfate (MTBS) as a conduction salt. It was found that densely packed films of [Cu_3_(BTC)_2_] crystallites with sizes of 2–50 μm could be easily grown by simple alterations of voltage and the composition of the electrolyte. Up to now, there are not practical applications of MOF membranes prepared by anodic deposition for gas separation and pervaporation.

**Figure 6 membranes-12-01205-f006:**
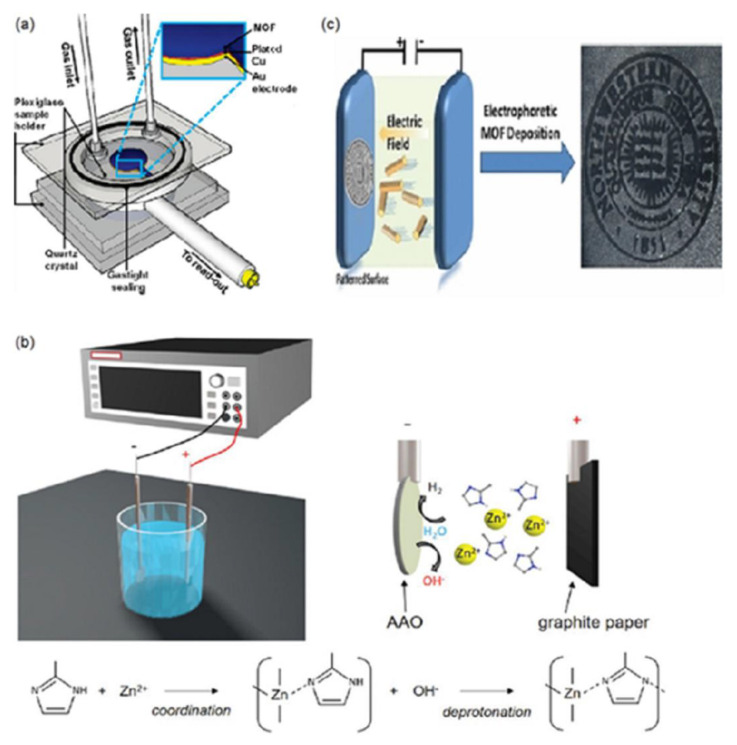
Illustration of electrodeposition of MOF membranes: (**a**) anodic deposition of [Cu_3_(BTC)_2_] membrane, (**b**) cathodic deposition of ZIF-8 membrane and (**c**) electrophoretic deposition of MOF membranes. Reprinted with permissions from Refs. [102,105,115].

A disadvantage of anodic dissolution is that the anode is necessarily corroded during the fabricating process of the MOF membrane, which can only apply to a few MOF membranes. To avoid this disadvantage, Dincă et al. illustrated that Zn_4_O(BDC)_3_ (MOF-5; BDC = 1,4-benzenedicarboxylate) was deposited at ambient temperature by cathodic electrodeposition [116]. Electrodes were used as chemically inert spectators that did not provide metal ions but provided electrons for the MOF membranes in the cathodic deposition process. To eliminate the potential negative effects on the environment caused by organic solvents, Wei and co-workers developed an aqueously cathodic deposition (ACD) approach to prepare the ZIF-8 membranes, adopting 100% water as the sole solvent instead of a supporting electrolyte or modulator [102]. In the electrolyte cell, anodic aluminium oxide (AAO) with the coating of a conductive layer was deployed as the working electrode, while the counter electrode was graphite paper without any pre-treatment. The two electrodes were parallelly separated by 1.5 cm and soaked in a ZIF-8 precursor solution and zinc cations were driven by electrical field towards the substrate to form a ZIF-8 membrane in only 60 min. The as-formed membrane demonstrated a best combination of 182 GPU permeance and 142 selectivity for the separation of C_3_H_6_/C_3_H_8_ mixtures.

Besides anodic dissolution and cathodic deposition methods, the electrophoretic deposition (EPD) method was also used to deposit the MOF membranes [105]. As-synthesized MOFs displaying net negative charges were first suspended in a toluene solution by sonication, and subsequently two identical fluorine-doped tin oxide (FTO) glass substrates as transparent conductive electrodes were placed in the MOF deposition solution. Partially charged MOF particles were finally driven towards the oppositely charged electrode to form MOF membranes under 90 V. In 2015, Zhu et al. deposited HKUST-1 membranes on porous stainless steel by the EPD method under 100 V for 120 s. In the single gas permeation experiments, the resultant HKUST-1 membrane exhibited the Knudsen separation factors 4.6 for H_2_/CO_2_, and 3.9 for H_2_/N_2_ [117].

#### 2.1.7. Other Methods

Recently, some other methods have been developed to fabricate continuous MOF membranes for gas separation and pervaporation. For example, Wang et al. utilized the melting properties of some MOFs including ZIF-62, ZIF-76, and TIF-4, to deposit the molten ZIF-62 phase on porous ceramic aluminate supports [118]. The molecular sieving ability of the as-synthesized ZIF-62 membrane was remarkably enhanced, giving a high separation factor up to 50.7 for H_2_/CH_4_, 34.5 for CO_2_/N_2_, and 36.6 for CO_2_/CH_4_. A new non-conventional approach for the fabrication of a continuous MOF membrane was reported [119]. In this method, a high pressure of 10,000 psi was used to press SIM-1 crystals into a wafer, and then the pressed wafer was treated with ethylenediamine to generate a free-standing SIM-1 membrane. The resulting SIM was used to separate water from water/ethanol mixtures at room temperature in pervaporation, exhibiting a flux of 460 g/m^2^·h with no ethanol detected in the permeate.

## 3. Application of Continuous MOF Membranes

### 3.1. Mechanisms for Gas Separation and Pervaporation

In gas separation and pervaporation, the solution-diffusion mechanism governs transportation permeance and selectivity of molecules across the membrane. Specifically, there are two factors (solubility and diffusivity) that play an important role in gas separation and pervaporation. The solubility of the targeted molecule relies on its interaction with the membrane material while the size of the targeted molecule, with respect to the window and pore sizes of the MOF, determines its diffusivity [14].

The solubility of the gas or liquid molecules is closely related to the equilibration between the gas or liquid molecules and the MOF, and therefore the adsorbent selectivity and adsorption capacity of the MOF play pivotal roles in solubility-based separations [120]. In this case, the membrane material will adsorb preferentially the targeted molecule while the window size of the MOF is larger than the kinetic diameters of all the gas or liquid molecules. If the desorption ability of the targeted molecules downstream is ideal enough to allow for faster diffusion of the targeted molecule, good separation results [14]. However, the solubility of the gas or liquid molecules on the MOF membrane is complicated because there is competition between limited adsorption sites as well as the potentials for cross-adsorbate interactions [120,121,122]. Recently, Matzger et al. presented a spectroscopic approach to accurately measure the adsorption behaviour in binary gas mixtures that will aid in the initial screening of adsorbents for specific applications [120].

In the diffusivity-based separation, the unique pore structures (window and pore sizes) of MOFs (Table 1) determine the separation performance. In this situation, the window size of the MOF falls between the target molecule and the other molecule. The targeted molecule can pass through the channels provided by the MOF while the other molecule will be blocked outside of the pores of the MOF. Notably, the frameworks of some MOFs are flexible, which will lead to the molecule with a kinetic diameter larger than the window size of the MOF passing through its corresponding membrane. In addition, the nonselective grain boundary effect and the inevitable linker rotation are also unneglectable factors [123].

### 3.2. Continuous MOF Membranes in Gas Separation

The intrinsic pore structures of MOFs have resulted in the surge of interest in developing continuous MOF membranes for hydrogen purification, CO_2_ removal, and propylene-propane separation. Table 2 summarizes the gas permeation properties of continuous MOF membranes fabricated by different methods. The H_2_/CO_2_, CO_2_/N_2_, CO_2_/CH_4_ and C_3_H_6_/C_3_H_8_ separation performance of continuous MOF membranes are drawn against the upper bounds in Figure 7.

**Table 2 membranes-12-01205-t002:** Continuous MOF membranes for gas separation.

MOF	Support	Method	Permeance (GPU)						Selectivity				Ref.
			H_2_	CO_2_	CH_4_	N_2_	C_3_H_6_	C_3_H_8_	H_2_/CO_2_	CO_2_/N_2_	CO_2_/CH_4_	C_3_H_6_/C_3_H_8_	
HKUST-1	N/A	in situ	4716.42	1149.25	1901.49	1402.99	N/A	N/A	4.10	0.82	0.60	N/A	[142]
HKUST-1	PVDF	in situ	6000	1044.78	1761.19	1313.43	N/A	N/A	5.74	0.80	0.59	N/A	[143]
MOF-5	alumina	secondary growth	2388.06	746.27	1164.18	895.52	N/A	N/A	3.20	0.83	0.64	N/A	[59]
ZIF-8	TiO_2_	in situ	179.1	38.81	14.93	14.93	N/A	N/A	4.61	2.60	2.60	N/A	[47]
ZIF-7	alumina	secondary growth	220.9	32.84	35.22	32.84	N/A	N/A	6.73	1.00	0.93	N/A	[144]
ZIF-90	Torlon^®^ fiber	secondary growth	567.16	310.45	208.96	89.55	N/A	N/A	1.83	3.47	1.49	N/A	[145]
ZIF-90	alumina	in situ	626.87	38.81	32.84	35.82	N/A	N/A	16.15	1.08	1.18	N/A	[146]
HKUST-1	Cu net	in situ	379.1	83.58	47.76	83.58	N/A	N/A	4.54	1.00	1.75	N/A	[53]
ZIF-8	alumina	in situ	N/A	N/A	N/A	N/A	24.15	0.59	N/A	N/A	N/A	41.00	[147]
[Ni_2_(L-asp)_2_(bipy)]	nickel mesh	secondary growth	3044.78	134.93	385.07	249.85	N/A	N/A	22.57	0.54	0.35	N/A	[148]
ZIF-8	Torlon	interfacial synthesis	1710.8	N/A	N/A	N/A	22.9	2.5	N/A	N/A	N/A	12.00	[77]
ZIF-8	alumina	Electrodeposition	5133	N/A	N/A	N/A	386	3.22	N/A	N/A	N/A	120.00	[149]
ZIF-90	alumina	in situ	850.75	37.61	8	N/A	N/A	N/A	22.62	N/A	4.70	N/A	[150]
ZIF-94	P84^®^	in situ	12.54	3.5	0.1	N/A	N/A	N/A	3.58	N/A	35.00	N/A	[151]
CAU-1	alumina	secondary growth	1515.5	3940.3	266.24	150.39	N/A	N/A	0.38	26.20	14.80	N/A	[152]
MIL-100	alumina	in situ	N/A	5283.58	4477.61	1462.69	N/A	N/A	N/A	3.61	1.18	N/A	[153]
UiO-66	alumina	Liquid phase epitaxy	313.43	1047.76	35.82	20.9	N/A	N/A	0.30	50.13	29.25	N/A	[154]
ZIF-62	alumina	melt-quenching	56.72	35.82	1.14	1.05	N/A	N/A	1.58	34.11	31.42	N/A	[118]
ZIF-8	alumina	interfacial synthesis	N/A	140.3	N/A	25.56	N/A	N/A	N/A	5.49	N/A	N/A	[74]
ZIF-8	nylon	interfacial synthesis	3373.13	N/A	N/A	746.27	N/A	N/A	N/A	N/A	N/A	N/A	[75]
ZIF-8	torlon	interfacial synthesis	N/A	22	N/A	0.42	N/A	N/A	N/A	52.38	N/A	N/A	[83]
ZIF-8	alumina	liquid-phase epitaxy	56.72	5.97	12.24	5.67	1.79	0.51	9.50	1.05	0.49	3.51	[85]
UiO-66-NH_2_	alumina	liquid-phase epitaxy	35,671.64	N/A	N/A	8268.66	N/A	N/A	N/A	N/A	N/A	N/A	[91]
ZIF-8	alumina	vapor deposition	N/A	N/A	N/A	N/A	479.4	6.48	N/A	N/A	N/A	73.98	[97]
ZIF-8	PVDF	vapor deposition	64,298.51	22,568.35	3592.1	4258.18	2498.94	56.16	2.85	5.30	6.28	44.50	[155]
ZIF-8	alumina	Electrodeposition	N/A	N/A	N/A	N/A	182	1.28	N/A	N/A	N/A	142.19	[102]

**Figure 7 membranes-12-01205-f007:**
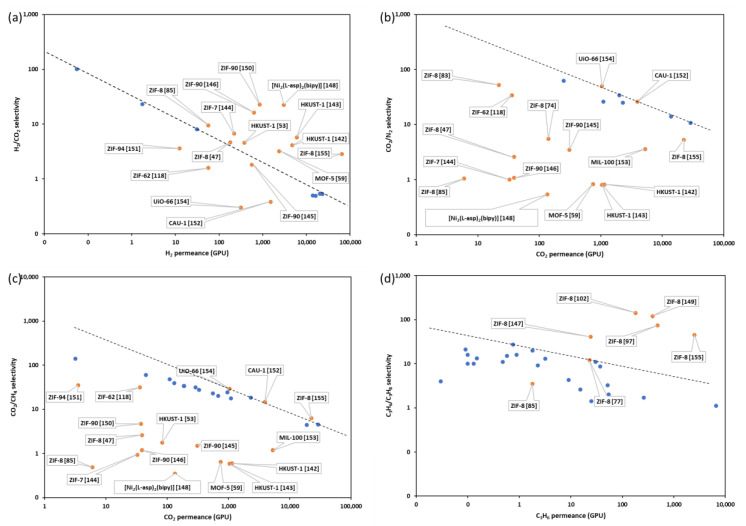
Selectivity-permeance comparisons against upper bounds [156,157]: (**a**) H_2_/CO_2_ separation, (**b**) CO_2_/N_2_ separation, (**c**) CO_2_/CH_4_ separation and (**d**) C_3_H_6_/C_3_H_8_ separation.

Continuous MOF membranes have mostly been used to act as molecular sieves to separate smaller molecules (H_2_, CO_2_, and C_3_H_6_) from larger molecules (CO_2_, N_2_, CH_4_, and C_3_H_6_). Excellent hydrogen permeance and selectivity over the upper bound is witnessed in Figure 7a because the electron-poor H_2_ is relatively smaller over CO_2_ molecules. However, the performance of the CO_2_ separation from N_2_ and CH_4_ using a continuous MOF membrane is below or close to the upper bound, though there is a large difference between the kinetic diameters of CO_2_ and N_2_ or CH_4_. Apart from the technical factors of membrane production, the window size of the MOF is another factor that might influence the separation performance of CO_2_ from N_2_ and CH_4_. From Table 1 and Table 3, we know the window sizes of ZIF-8 and MIL-96 fall exactly between the kinetic diameters of CO_2_ and N_2_ or CH_4_. However, it should be noted that some MOFs have a flexible nature and dynamic properties, like the effective aperture size of ZIF-8 between 4.0 to 4.2 Å [158]. That means, the effective aperture sizes of most MOFs in Table 1 are larger than the kinetic diameters of CO_2_, N_2_ and CH_4_, resulting in that most continuous MOF membranes cannot exhibit sharp molecular sieving. Furthermore, the interaction, including polarizability, magnetic susceptibility, permanent dipole moment, quadrupole moment, etc. between the gas molecule and the MOF membrane needs to be highly emphasized [159]. From the membrane material aspect, the strong polarizing groups from the MOF ligands such as 1,4-benzenedicarboxylate and imidazolate-2-carboxyaldehyde, and high-valence metal precursors such as Zr^4+^, Cr^3+^, Al^3+^, and Fe^3+^ can make MOF membrane show high adsorption to the gas molecule with a great permanent quadrupole moment, like CO_2_, but strong adsorption can also result in low diffusion coefficients according to recent research [159,160,161]. Promising molecular sieve performances for the separation of C_3_H_6_ and C_3_H_8_ have been witnessed using continuous MOF membranes (Figure 7d). Although reports of C_3_H_6_/C_3_H_8_ separation have been limited to ZIF-8, different fabrication methods can lead to different separation performance, such as ZIF-8 [72] and ZIF-8 [142]; even different membrane supports using the same membrane fabrication method can have different separation performance, such as ZIF-8 [92] and ZIF-8 [150].

### 3.3. Continuous MOF Membranes in Pervaporation

There are few reports on the continuous MOF membrane being applied in pervaporation, though many continuous MOF membranes have been successfully used for gas separation. In addition to the high requirements for the integrity of the membrane, there are also higher chemical-stability requirements for the membrane, considering that pervaporation is a liquid-phase process. With continuous developments in the field of MOF-based membranes, increasing attention has been paid to the application of continuous MOF membranes in pervaporation. In 2021, Luis et al. published a review on MOF-based membranes in pervaporation [9]. In this review, the design strategies, developments, and performance of continuous MOF membranes were discussed in detail. Compared to other types of membranes, especially polymeric membranes, continuous MOF membranes showed encouraging performance (Table 4 and Figure 8). In the dehydration of organics, SIM membranes exhibited the highest selectivity for the dehydration of EtOH. This can be attributed to the small window of the SIM. The Ni_2_(l-asp)_2_bipy and UiO-66 had higher water permeances with moderate selectivities because of their larger window sizes. In addition, the hydrophilic surface of the Ni_2_(l-asp)_2_bipy membrane also contributed to this remarkably high water permeance. In the recovery of organics, the hydrophobicity of ZIF-71 did not endow its corresponding membranes with EtOH selectivity because the small difference between the window size of ZIF-71 and the kinetic diameter of EtOH (Table 5). In the separation of organic–organic mixtures, the window size of UiO-66 (0.60 nm) is far larger than the kinetic diameter of MeOH (0.38 nm), close to that of MTBE (0.62 nm), thus the UiO-66 membrane exhibited the largest MeOH permeance with a moderate selectivity in contrast to the other types of membranes. As it is the same series of the MOF as UiO-66, the polarity of UiO-66-NH_2_ is switched from hydrophobicity of UiO-66 to hydrophilicity by grafting amino groups on the organic ligand. The change in the polarity greatly facilitates the adsorption of strongly-polar thiophene [9]. In addition, the kinetic diameter of thiophene (0.46 nm) is much smaller than the window size of UiO-66-NH_2_ (0.75 nm), thus it can spread across the membrane by molecular diffusion. That is the reason why the UiO-66-NH_2_ membrane showed the highest thiophene permeance and selectivity.

## 4. Conclusions and Perspectives

In gas separation and pervaporation, continuous MOF membranes have exhibited promising potentials in pushing the polymer upper bounds. The increasing progress in the continuous MOF membrane is attracting increasing attention. The following issues are recommended to pay attention to in future research: (1) Exploring the membrane fabrication techniques to obtain defect-free continuous MOF membranes; (2) optimizing membrane structures including fabrication parameters, support membrane, and crystal nucleation and growth; (3) characterizing adsorption and desorption behaviours of the gas or liquid molecules accurately in the MOF using effective techniques before practical applications; (4) measuring the effective window size of the MOF for more efficient separations; (5) selecting the suitable MOF for the target application by computational simulation and calculations, not limited to ZIFs; (6) paying more attention to the application of continuous MOF membranes in pervaporation. Although the application of continuous MOF membranes for gas separation and pervaporation is still in its infancy, we believe that exciting breakthroughs are on the horizon.

## Figures and Tables

**Figure 1 membranes-12-01205-f001:**
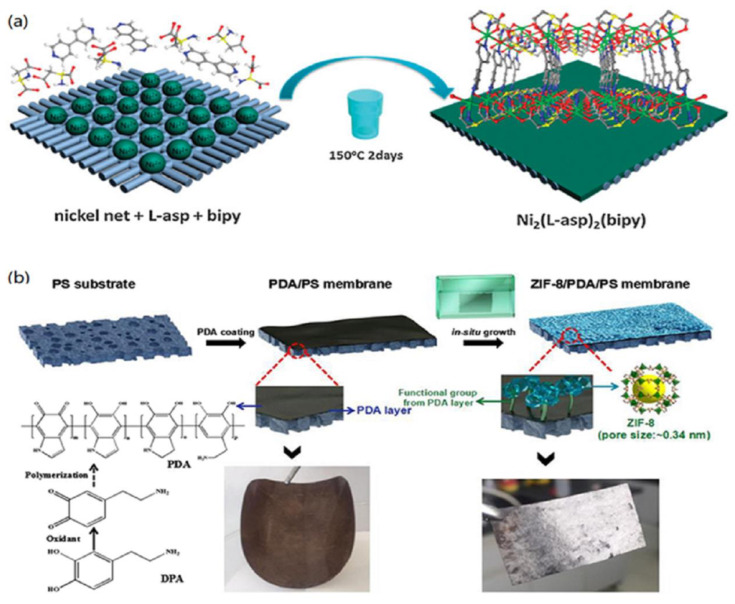
Schematic illustrations of the in-situ synthesis of continuous MOF membranes: (**a**) the growth of Ni_2_(L-asp)_2_(bipy) on an unmodified nickel support and (**b**) the growth of ZIF-8 membrane on a modified PS support. Reprinted with permission from Refs. [12,40].

**Figure 8 membranes-12-01205-f008:**
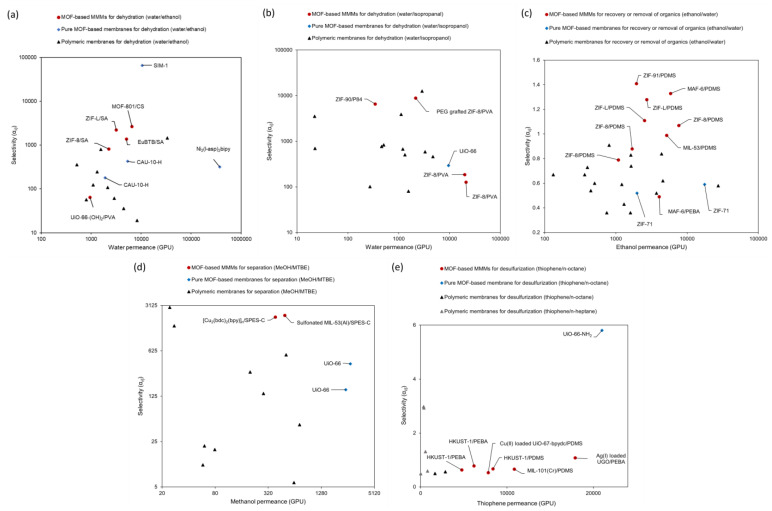
Separation performance of continuous MOF membranes in pervaporation: (**a**) H_2_O/EtOH separation, (**b**) H_2_O/Isopropanol, (**c**) EtOH/H_2_O, (**d**) MeOH/MTBE and (**e**) Thiophene/n-octane or n-butane. Reprinted with permission from Ref. [6].

**Table 1 membranes-12-01205-t001:** Brief overview of structural parameters of MOFs utilized in gas separation and pervaporation.

MOF Name	Metal	Ligand	Formula Composition	Pore Volume (cm^3^/g)	BET Surface Area (N_2_) (m^2^/g)	Window Size (Å)	Ref.
UiO-66	Zr	1,4-benzenedicarboxylate (BDC)	Zr_6_O_4_(OH)_4_(BDC)_6_	0.36	970	6	[124]
MOF-5 (Zn)	Zn	BDC	Zn_4_O(BDC)_3_	1.4	3800	12	[125]
ZIF-71	Zn	4,5-dichloroimidazole (dcIm)	Zn(dcIm)_2_	0.452	1007	4.8	[126]
CAU-10-H	Al	benzene-1,3-dicarboxylate (1,3-H_2_BDC)	[Al(OH)(benzene-1,3-dicarboxylate)]·nH_2_O	0.43	635	7	[127]
ZIF-7	Zn	Benzimidazole (H-bIM)	Zn(bIM)_2_	0.207	380	3	[128]
ZIF-8	Zn	2-Methylimidazole (Hmim)	Zn(mim)_2_	0.554	1344	3.4	[126]
Sm-DOBDC	Sm	2,5-dihydroxy-1,4-benzenedicarboxylate (DOBDC)	Sm_6_(OH)_8_(DOBDC)_6_	0.263	520	N/A	[129]
SIM-1(ZIF-94)	Zn	4,5-imidazolecarboxaldehyde	Zn(4-methyl-5-imidazolcarboxaldehyde)_2_	0.19	471	2.6	[130,131]
CAU-1	Al	2-amino-1,4-benzene dicarboxylic acid (H_2_BDC-NH_2_)	Al_8_(OH)_4_(OCH_3_)_8_	0.59	1021.7	4	[132]
MIL-100	In	1,3,5-benzenetricarboxylic acid (H_3_BTC)	In_3_O(H_2_O)_2_OH(BTC)_2_	0.636	1456	4.6 and 8.2	[133]
ZIF-62	Zn	Imidazole (Im) and benzimidazole (Bim)	Zn(Im)_1.75_(Bim)_0.25_	0.20	257	1.4	[134,135]
Ni_2_(L-asp)_2_(bipy)	Ni	L-aspartic acid and 4,4′-bipyridine (L-asp and bipy)	Ni_2_(L-asp)_2_(bipy)	N/A	247 (CO_2_)	3.8 Å × 4.7 Å	[136]
ZIF-90	Zn	imidazolate-2-carboxyaldehyde (ICA)	Zn(ICA)_2_	0.561	1360	2.86	[126]
UiO-66-NH_2_	Zr	2-amino-1,4-benzenedicarboxylic acid (H_2_BDC-NH_2_)	Zr_6_O_4_(OH)_4_(BDC-NH_2_)_6_	0.48	1073	7.5	[137,138]
MIL-96	Al	H_3_BTC	Al_12_O(OH)_18_(H_2_O)_3_(Al_2_(OH)_4_) [BTC]_6_·24H_2_O	0.24	629.98	3.6 Å × 4.5 Å	[139,140]
HKUST-1	Cu	benzene 1,3,5-tricarboxylate (BTC)	Cu_3_(BTC)_2_(H_2_O)_3_	0.75	1663	9	[141]

**Table 3 membranes-12-01205-t003:** Kinetic diameter for various molecules.

Molecule	Kinetic Diameter (Å)
H_2_	2.89
CO_2_	3.30
N_2_	3.64
CH_4_	3.80
C_3_H_6_	3.90
C_3_H_8_	4.30

**Table 4 membranes-12-01205-t004:** Continuous MOF membranes for pervaporation.

MOF	Support	Method	Mixture (i/j)	Mass Ratio (i/j)	T (°C)	Flux (g/m^2^·h)	i Permeance (GPU)	j Permeance (GPU)	Separation Factor (β_i/j_)	Selectivity (α_i/j_)	Ref.
CAU-10-H	alumina	secondary growth	H_2_O/EtOH	10–90	40	397	5391.77	12.52	324	430.61	[162]
CAU-10-H	alumina	secondary growth	H_2_O/EtOH	10–91	65	493	1914.48	10.63	148	180.09	[162]
Ni_2_(l-asp)_2_bipy	SiO_2_	secondary growth	H_2_O/EtOH	50/50	30	28,100	373,279.57	1169.2	73.6	319.26	[65]
Sm-DOBDC	alumina	secondary growth	H_2_O/EtOH	5/95	50	786.4	N/A	N/A	>9481	N/A	[129]
SIM-1	glass	mechanical press	H_2_O/EtOH	55.82/44.18	25	460	10,493.15	0.1589	>10,000	>66,036.24	[119]
UiO-66	Yttria-Stabilized Zirconia	in situ	H_2_O/EtOH	5/95	50	2550	8776.12	276.85	32.9	31.7	[163]
MIL-96	alumina	secondary growth	H_2_O/Ethyl acetate	4.4/95.6	60	70	N/A	N/A	>1279	N/A	[61]
UiO-66	alumina	liquid-phase epitaxy	EtOH/H_2_O	10/90	50	1490	N/A	N/A	4.9	N/A	[164]
ZIF-71	alumina	interfacial synthesis	EtOH/H_2_O	5–95	25	2601	17,619.83	29,951.78	6.88	0.59	[76]
ZIF-71	ZnO	secondary growth	EtOH/H_2_O	5–95	25	322.18	1980.52	3833.32	6.07	0.52	[165]
UiO-66	alumina	secondary growth	MeOH/MTBE	5–95	40	1210	2733.36	6.92	597	395.06	[166]
UiO-66	alumina	secondary growth	MeOH/MTBE	15/85	40	1920	2423.83	15.35	147	157.9	[166]
UiO-66-NH_2_	alumina	in situ	Thiophene/n-octane	0.13/99.87	40	2160	~21,000	~4000	17.96 (enrichment factor)	~5.8	[167]

**Table 5 membranes-12-01205-t005:** Physical parameters of liquid solvents in pervaporation.

Solvent	Kinetic Diameter/Å
Water	2.65
Methanol (MeOH)	3.8
Ethanol (EtOH)	4.3
Isopropanol	4.6
Thiophene	4.6
Methyl tert-butyl ether (MTBE)	6.2
Ethyl acetate	5.2
N-octane	4.3

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
