# Peer review of "Recent Advances in Continuous MOF Membranes for Gas Separation and Pervaporation"

_membranes, 2022, doi:10.3390/membranes12121205_

Round 1

Reviewer 1 Report

Manuscript No.: membranes-2036704

Title: Recent advances in pure MOF membranes for gas separation and pervaporation

This review article covers the topic of the synthesis pathway and separation performance of pure MOF membranes. In my opinion, article presents valuable content. The subject and as-summarized information regarding MOF’s preparation as well as corresponding separation capability are informative. However, some modifications should be considered.

Please see below more specific remarks:

1.        The Introduction section should involve the motivation for applying the membrane-based process in gas separation or organic liquid purification to showcase the importance of effective and efficient separation process to environment.

2.        The comparison of diversity synthesis pathways such as in-situ growth should be incorporated into the revised version, based on their pros/cons and energy consumption.

3.        Which one transportation mechanism through MOF membrane is pivotal? Molecular sieving or solution-diffusion

4.        Since MOF membrane falls into thin film, the adopted gas permeation unit should be Barrer, instead of GPU, so as to neglect the influence of membrane thickness.

5.        After reviewing a series of separation performance of pure MOF membranes, the authors should guide the suitable MOF membrane for specific applications such as carbon capture, hydrogen recovery, and paraffin separation.

6.        The window size of UiO-66 is 6 angstrom, while its CO2/N2 and CO2/CH4 performance exceeds the upper bound. Please explain the reason of the counterintuitive phenomenon.

7.        Figure 7 and 8- please improve the text. In the current version, they are illegible.

Reviewer 2 Report

Based on the application of gas separation and pervaporation, this manuscript comprehensively introduces the preparation methods of pure MOF membranes, including physical and chemical processes. In addition, the performances of pure MOF membranes are introduced in the form of tables. In general, the manuscript makes a detailed statistics of realm-related works, and there is a reasonable basis for classifying film-forming processes. However, there are still some issues to be addressed.

1.      In the title, authors emphysized on the topic of pure MOF membranes. However, in the manuscript, most of the MOFs are grew on substrates and applied as composites. Therefore, this reviewer would suggest authors to revise the expression of pure MOF membranes.

2.      The bulk and center of this review is actually about the preparation of pure MOF films, and this should be pointed out in the introduction section.

3.      As a review article, comprehensive literature searching is necessary. Some important, relevant and recent articles about the structure, properties and applications of MOFs should be included in proper places: Porous Hafnium-Containing Acid/Base Bifunctional Catalysts for Efficient Upgrading of Bio-Derived Aldehydes; A poly(amidoxime)-modified MOF macroporous membrane for high-efficient uranium extraction from seawater; Hierarchical porous Co3O4 nanocages with elaborate microstructures derived from ZIF-67 toward lithium storage; etc.

4.      Please standardize the size and font of the captions in each figure.

5.      Characterizing the performance of adsorbents toward mixtures of gases is critical in most adsorptive separations, herein, the new technologies are suggested to attach importance to, referring: Microscale Determination of Binary Gas Adsorption Isotherms in MOFs.

6.      The mechanism of MOFs for gas separation and permeation is mentioned in the text, but in insufficient detail. A properly developed presentation and corresponding graphical presentation would make this issue more accessible to the reader.

7.      The growth of MOF membranes on biomass materials such as wood does not receive much attention in the paper, which should be highlighted also in the work with supporting articles: Chemical Engineering Journal 446, 136851, 2022; Chem 8 (9), 2342-2361, 2022.

8.      Some of the figures should be modified to have a better resolution and readability, especially the texts in the figures.  

9.      More perspectives including the challenges and possible solutions should be provided to guide the future research.

10.  Please carefully check the whole manuscript. There are still some typos and grammar issues. In addition, please carefully check the references to ensure the full information is included.

11.  When given the references, all the author names should be listed.

Round 2

Reviewer 2 Report

Accept in present form